# A Haplotype GWAS in Syndromic Familial Colorectal Cancer

**DOI:** 10.3390/ijms26020817

**Published:** 2025-01-19

**Authors:** Litika Vermani, Johanna Samola Winnberg, Wen Liu, Veronika Soller, Tilde Sjödin, Mats Lindblad, Annika Lindblom

**Affiliations:** 1Department of Molecular Medicine and Surgery, Karolinska Institutet, 17176 Stockholm, Sweden; wen.liu@neuro.uu.se (W.L.); veronika.soller@gmail.com (V.S.); 03tildesjodin@gmail.com (T.S.); 2Division of Surgery, Department of Clinical Science Intervention and Technology (CLINTEC), Karolinska Institutet, 17177 Stockholm, Sweden; johanna.samola.winnberg@ki.se (J.S.W.); mats.lindblad@ki.se (M.L.); 3Department of Upper Abdominal Diseases, Karolinska University Hospital, 17177 Stockholm, Sweden; 4Department of Neuroscience, Uppsala University, 75237 Uppsala, Sweden; 5Department of Clinical Genetics and Genomics, Karolinska University Hospital, 17176 Stockholm, Sweden

**Keywords:** GWAS, colorectal cancer, gastric cancer, prostate cancer, familial, genetic, exome sequencing, genome sequencing

## Abstract

A previous genome-wide association study (GWAS) in colorectal cancer (CRC) patients with gastric and/or prostate cancer in their families suggested genetic loci with a shared risk for these three cancers. A second haplotype GWAS was undertaken in the same colorectal cancer patients and different controls with the aim of confirming the result and finding novel loci. The haplotype GWAS analysis involved 685 patients with colorectal cancer cases and 1642 healthy controls from Sweden. A logistic regression model was used with a sliding window haplotype approach. Whole-genome and exome sequencing datawere used to find candidate SNPs to be tested in a nested case-control study. In the analysis of 685 colorectal cancer cases and 1642 controls, all ten candidate loci from the previous study were confirmed. Fifty candidate loci were suggested with a *p*-value < 5 × 10^−6^ and odds ratios between 1.35–6.52. Two of the 50 loci, on 13q33.3 and 16q23.3, were the same as in the previous study. Whole-genome or exome data from 122 colorectal cancer patients was used to search for candidate variants in these 50 loci. A nested case-control study was performed to test genetic variants at 11 loci in a cohort of 827 familial colorectal cancer and a sub-cohort of 293 familial CRC cases with colorectal, gastric, and/or prostate cancer within their families and 1530 healthy controls. One SNP, rs115943733 on 10q11.21, reached statistical significance (OR = 3.26, *p* = 0.009). Seven SNPs in 4 loci had a higher OR in the smaller cohort compared to the larger study CRC cases. The results in this GWAS gave support for suggested loci with an increased shared risk of CRC, gastric, and/or prostate cancer. Further studies are needed to confirm the shared risk to be able to use this information in cancer prevention.

## 1. Introduction

Colorectal cancer is the third most common cancer and the second most common cause of cancer death worldwide [1]. Approximately 75% of patients diagnosed with CRC exhibit sporadic disease, showing no apparent signs of inheriting the disorder. In contrast, 10% to 30% of patients have a family history of CRC, indicating a potential hereditary component, shared environmental exposures, common risk factors among family members, or a combination thereof [2]. Pathogenic variants in high-penetrance genes have been identified as the underlying cause of inherited cancer risk in certain families predisposed to CRC. However, these variants are estimated to account for only around 5% of CRC cases overall [3]. A known CRC syndrome, Lynch syndrome, with a genetic predisposition to non-polyposis CRC and other cancers, is caused by germline disease-causing variants in the DNA mismatch repair (MMR) genes *MLH1*, *MSH2*, *MSH6*, *PMS2*, or *EPCAM* [4]. Familial Adenomatous Polyposis (FAP) is caused by pathogenic variants in *APC* or *MUTYH* (MAP) [4]. However, despite these advances, a significant proportion of familial cases remain genetically uncharacterized [5].

Today, colorectal cancer and many other cancer types and common diseases are suggested as complex diseases caused by a combination of predisposing genetic variants in low-risk genes and environmental factors [6]. Thus, instead of searching for single high-penetrant genes, Genome-Wide Association Studies (GWAS) have been used to identify common genetic loci associated with an increased risk of and other cancers, as well as several common disorders. The findings have been validated using studies of thousands of individuals [7,8]. Many GWAS for risk in CRC have been performed to date and have identified more than 205 independent risk associations [8]. Still, these cannot explain most of the familial increased CRC risk. Studies examining pleiotropy across multiple cancer-associated loci have been done to identify common mechanisms of cancer development and progression [9].

A new CRC syndrome was suggested based on previous findings in families with colorectal and other cancers, most importantly, gastric and prostate cancers [10]. Linkage analysis in families with colorectal, gastric, and prostate cancer was undertaken to study the pattern of inheritance in the underlying hypothetical syndrome [11]. This analysis did not suggest any high penetrant disease-associated locus; instead, a complex disease for this putative syndrome was suggested [11]. To test this hypothesis, a haplotype genome-wide association study (GWAS) was first conducted, involving CRC patients who had gastric and/or prostate cancer among their relatives. The study used healthy, unrelated Swedish twins as controls [12]. The results suggested 10 risk loci associated with increased risk of colorectal, possibly also gastric, and prostate cancer [12]. Since another cohort of controls was available, it was possible to test the hypothesis further in a new study using the same CRC cases, but with this smaller cohort of geographically better-matched control persons recruited at the same time as cases and genotyped with the same chip [10,13,14]. The aim of the new study was to replicate the ten loci found in the previous study and to search for additional new loci for the same suggested syndrome.

## 2. Results

### 2.1. Replication of the Loci Found in Previous Haplotype GWAS

A test utilizing the 10 loci found in the previous study was undertaken. The positions for each of the ten statistically significant haplotype loci were searched for among the results from the new analysis. Two identical haplotypes, on loci 1q32.2 and 3q29, were found (Table 1). The exact same haplotypes could not be identified for any of the other eight loci because the controls used in this study were genotyped with a different chip than controls in the previous study, thus leading to somewhat different haplotypes in the new analysis. However, statistically significant findings of haplotypes at the same loci were still found (Table 1). Nine of the ten loci were confirmed with ORs > 1 and statistical significance *p* < 0.005). One locus, 16q23.3, had an OR = 3.6 and *p* = 0.06 (Table 1). The odds ratio (OR) in this new study using few but better-matched controls was higher in four and lower in six of the tested loci as compared to the previous findings (Table 1).

### 2.2. Haplotype GWAS with Correct Cases and Better-Matched Controls

Next, a sliding window (1–25) haplotype GWAS was conducted using 685 CRC cases and 1642 healthy controls (Appendix A). As the cohort of controls was smaller than the one used in the previous study, loci reaching *p* < 5 × 10^−6^ were considered for follow-up in this study. Only one locus on chromosome 9p24.3 reached a *p*-value generally considered statically significant in GWAS (OR = 2.3, *p* = 2.41 × 10^−8^) [15]. A total of fifty-five haplotypes at 50 loci had *p* < 5 × 10^−6^ and ORs from 1.35 to 6.52 (Table 2). Thirty-six loci had 1–3 coding genes in the haplotype region, and many had non-coding genes, RNA genes, pseudogenes, or other regulatory elements between the first and last position at the locus. Since loci with other elements except coding genes could be equally important, a search for candidate SNPs was undertaken in all fifty-five haplotype regions. Two loci among the 55, one on 13q33.3 and one on 16q23.3, were found also in the first study [12]. The OR for locus 13q33.3 (*MYO16*) was higher in the present study (Table 2). The other locus found in both studies was 16q23.3 (*CDH13*). However, the *CDH13* locus involved two different regions. The region identified in the previous study was 82871769–82899877 (Table 1), and the region in this study was 82673410–82691564 (Table 2). Both haplotypes were within the gene *CDH13* and suggested two unique risk variants in this gene.

### 2.3. Selection of Candidate Variants for SNP Analysis

Next, an attempt was made to find gene variants in each of the 55 haplotypes at all 50 loci. Variants in each locus are candidates to act as causative of an increased risk at that locus. In total, 122 familial CRC cases with gastric and/or prostate cancer in their families were sequenced using WES or WGS, and the data was searched for candidate SNPs in all 55 haplotype regions. Altogether, 33 variants at 14 loci (2p23.1, 2q33.1, 4q31.1, 4q31.3, 4q35.1, 6q25.3, 6q27, 10q11.21, 11q14.2, 13q12.11, 13q12.3, 14q32.11, 18q21.2, and 20q13.33) were selected and set up for testing. The variants were genotyped in MALDI-TOF analysis. As some SNPs were not compatible with each other in the primer design, they were eliminated, and after further QC, 17 variants in 11 loci were subject to testing. The 11 loci were 2q33.1, 4q31.1, 4q31.3, 4q35.1, 6q27, 10q11.21, 11q14.2, 13q12.11, 13q12.3, 14q32.11, and 20q13.33, and the 17 SNPs were rs114046582, rs115872046, rs115943733, rs1213626528, rs139227237, rs140852957, rs141501417, rs2427307, rs34803482, rs34851370, rs35516773, rs358314, rs358326, rs5743708, rs61735304, rs7150480, and rs72716373.

Finally, a case-control study tested all 17 candidate genetic variants in 827 familial CRC cases and 1530 healthy control persons. Among the cases, 293 also had a family history of gastric and/or prostate cancer. None of the loci tested in the association analysis showed a statistically significant result, although OR was >1 for nine of the tested SNPs (Table 3). A second association study was performed using a sub-cohort of only the 293 familial CRC cases with colorectal, gastric, and/or prostate cancer within their families and the same 1530 controls. In this smaller cohort, one SNP, rs115943733 on 10q11.21 (TMEM72), reached statistical significance (OR = 3.26, *p* = 0.009) (Table 3). Seven SNPs in 4 loci had a higher OR as compared with the previous analysis of 827 familial CRC cases, giving some support to the hypothesis of a higher risk of cancer in the smaller cohort of CRC cases selected because of their family history also including gastric and/or prostate cancer (Table 3).

## 3. Discussion

This haplotype GWAS study on the hereditary risk of colorectal, gastric, and prostate cancer replicates and confirms ten previously found candidate loci and suggests 50 novel candidate loci. A previous GWAS from the group on familial CRC cases with gastric and/or prostate cancer in their families used healthy controls from the Swedish twin registry. That study suggested ten risk loci with OR 1.71–3.62 and *p*-values of 2.56 × 10^−10^–4.47 × 10^−8^ [12]. The present study used a control population from the same geographical region as the cases, hoping to confirm and improve the results with better-matched controls. Cases and controls were in this study genotyped with the same chip, while, in the previous study, the controls were genotyped with a different chip, which could affect the results. The number of the new controls was smaller, because this time, the focus was on the odds ratio rather than the *p*-value to compare the results. Because different chips were used for controls in this and the previous study, generated haplotypes could be different because exact haplotypes could not always be compared. Instead, the positions for the chromosomal loci were considered, regardless of the actual haplotypes. The use of different SNPs in a GWAS means that a locus might be missed, but new loci could also be detected. Thus, the lack of confirmation of results from a previous study when replication used different SNPs for analysis does not rule out previous findings. Similar to the previous study using haplotype analysis, ORs were much higher compared to what has been seen in single-variant GWASs. This is probably because haplotype GWAS identifies rare loci, while single-variant GWAS finds common loci, and rare loci will only be found if they are associated with a higher OR and risk. Which *p*-values will be used can be discussed. In this study, a less stringent *p*-value was used; it could be argued that this is not correct. However, the finding of all these genes in the loci certainly gives strength to the results.

A first comparison of the 10 loci identified in the previous study, with findings in this study, noted that all 10 were confirmed with statistically significant positive ORs and statistically significant results, except the locus on chromosome 16q23.3 and the gene *CDH13*, which was only borderline significant at the exact positions (Table 1). However, another haplotype region in the same gene was observed with OR = 6.35, *p* = 3.18 × 10^−6^, and the same locus/gene (Table 2). This means that two risk variants in the same gene were suggested from our combined results, one between base pair position 82871769 and 82899877 in the previous study (Table 1) and one between the positions 82673410 and 82691564 in this new study with other controls (Table 2). The ORs were different. In the first study, the OR was 3.60 for the first haplotype in *CDH13*, which was 1.54 for the same positions in this study. The other sub-locus in the *CDH13* had, in this study, an OR of 6.35, suggesting that these two variants were possibly associated with different risks. However, it could also be explained by the fact that the two different control populations used different SNPs, different numbers, and came from different geographical regions. Once the targeting variant at these two sub-loci is identified, this will be explained. The other locus, confirmed among the best 10 loci in this study, was *MYO16* on 13q33.3 (OR = 2.54, *p* = 1.86 × 10^−6^). Here, this locus had a higher OR, 2.54, vs 1.71 in the previous study. The haplotype region in this study (109796718–109897922) was bigger than the one found in the first study (109832287–109897922). One candidate variant in *MYO16* was tested in the previous study and had a positive OR = 1.24, supporting an increased risk of CRC and possibly also gastric and/or prostate cancer. The finding of a positive association between a certain variant and risk does not prove that the variant is causative—but it is a candidate for further testing in bigger association studies. The genes involved in these ten loci and their possible role in CRC were previously discussed [12].

This study found one locus with a *p*-value generally considered statistically significant for GWAS on 9p24.3 (OR = 2.3, *p* = 2.41 × 10^−8^). No gene was suggested at this locus. We still searched for SNPs at all 50 loci, which reached a *p*-value < 5 × 10^−6^ for further study. The ORs observed in our haplotype GWAS were higher (ranging from 1.35 to 6.52) compared to typical GWAS findings, which mostly stay below OR = 2 [16,17,18,19,20]. This is the same as we have found also in previous haplotype GWAS [14]. In the 50 loci, 46 genes were found, almost all implicated in cancer already (Table 2). Many of the genes, *E2F6*, *TLR2*, *SFRP2*, *PIK3R1*, *EYA4*, *THBS2*, *ELMO1*, *RELN*, *GAS1*, *FZD4*, *KLF12*, *TDP1*, *SCG5*, *CDH13*, and *TCF4*, have been published in relation to cancer more than 50 times, and some have been published in hundreds or even thousands of publications in relation to cancer and/or CRC (Table 2). A detailed report of all putative genes involved is outside the scope of this paper. However, some of the most studied genes in relation to cancer, and in particular CRC, will be mentioned briefly. One paper found that *RBM15*, by regulating E2F2, boosted malignant processes in CRC [21]. Inflammation is a known risk factor in cancer, including CRC. Toll-like receptors (TLRs) are known players in inflammatory response. A low expression of *TLR2* in normal mucosa suggested that normal mucosa might contribute to a systemic inflammatory response [22]. Abnormal methylation of secreted frizzled-related proteins (SFRPs), specifically *SFRP2*, has been associated with cancer risk, especially in hepatocellular and colorectal carcinoma [23]. The same gene, *SFRP2*, as well as another gene suggested in Table 1, *EYA4*, were mentioned in relation to CRC risk and the effect of microbiota composition in methylation [24]. The phosphatidylinositol 3-kinase signaling pathway is known to be involved in cancer, and *PIK3R1* has shown association with risk in CRC [25]. Hub genes are known to be involved in signal transduction and metabolic pathways in cancer. The gene *THBS2* was shown to be implicated in metastatic CRC [26]. *ELMO1* was shown to mediate tumor progression in CRC [27]. Reelin, an extracellular matrix protein, is known to reduce the susceptibility to dextran sulfate sodium (DSS)-colitis. The gene has also been suggested to possibly protect against CRC by maintaining intestinal epithelial cell homeostasis and providing resistance against colon pathology [28]. The TCF family of genes is known to regulate Wnt/B-catenin target gene expression. The *TCF7L1* gene was shown to regulate CRC cell migration by repressing *GAS1* expression [29]. One study showed that overexpression of *FZD4* in CRC modified wnt/B-catenin signaling in CRC cells [30]. Krüppel-like factors have been suggested in malignancy by regulating various cellular functions. One study of these functions found a dramatic downregulation of KLF genes, including *KLF12*, in CRC [31]. Top1/Tdp1 are mediators of a DNA damage repair pathway and can influence the risk of CRC [32]. Polygenic modeling identified 10 SNP in the *SCL22A3*, *SCG5*, *GREM1*, and *STXBP5-AS1* genes in an Indonesian CRC GWAS [33]. *CDH13* is known to be downregulated in various cancers, and *CDH13* promotor methylation has been suggested to act in CRC initiation and progression [34]. Mutations in the Wnt components, specifically *TCF4*, were found in microsatellite-unstable colon cancers [35].

The aim of this study was to replicate the loci from our previous study and to possibly find more loci with an increased risk of CRC, gastric cancer, and prostate cancer. A search of putative candidate SNPs in these loci was done in the same way as for the ten loci in the previous study. The same sequenced data from 122 patients was used to find markers for testing. In total, 33 SNPs were found available for genotyping and, finally, 17 SNPs could be used in two association studies. The first used 827 familial CRC cases and 1530 controls, and the second used a sub-cohort of 293 familial CRC cases with a family history also including gastric and/or prostate cancer in close family members and the same controls. Only 11 of the 50 loci could be tested for one to four markers and only one variant was found to be statistically significant (OR = 3.26, *p* = 0.009). The first analysis of 827 familial CRC cases versus 1530 healthy controls found support with ORs > 1 for seven of the 11 tested loci. When the analysis was redone in the smaller cohort of 293 familial CRC selected for the colon-gastric-prostate familiality, six loci had OR > 1, with three of them having higher OR and one having statistically significant results. This is interpreted as a support of those loci. However, too few of the loci could be tested for specific gene variants and none is therefore ruled out. The fact that so many of the suggested 50 haplotype regions involved coding genes, and that most of them already had been implicated in cancer, including CRC, can also be seen as supportive of the regions suggested as risk loci in this study.

The major limitation of both the previous and this study was that too few patients were sequenced. Since almost all suggested haplotype regions (loci) had an estimated sample frequency below 10%, the chance of finding the putative haplotype target gene variants in only 122 patients was small. Besides, even if ORs in haplotype GWAS are higher compared to those in single-variant GWAS, the numbers of cases and controls were too small to give statistically significant results. Another limitation was that the same cases were used. However, after careful consideration, it was decided to undertake this study since the extra control persons were available, although other cases were not. Perhaps the choice of patients for sequencing was not the best either, since familial CRC cases were used, while in the first GWAS, mostly sporadic cases (with a family history of gastric and/or prostate cancer) were used. Still, the study could replicate the ten loci from the first study, finding new candidate loci with high OR, although not statistically significant in this smaller study. Thus, both this and the previous study found risk loci were suggested to harbor genetic variants associated with a risk of colon, and possibly also gastric and prostate, cancer. Moreover, most of the genes found in our studies were already implicated in various cancers. The current study suggested that geographically better-matched controls resulted in better risk stratification as the odds ratio for the respective loci was higher, even though the cohort was smaller, and the results did not reach statistical significance. The study can serve as a proof of principle to find candidate risk markers in loci from haplotype GWAS and demonstrate the importance of the selection of population and size of the experiment in numbers to be able to give statistically significant results.

## 4. Materials and Methods

### 4.1. Cases and Controls for GWAS

Cases were selected as a part of a multi-center study, the Colorectal Cancer Low-risk study, with newly diagnosed CRC patients from the middle of Sweden between 2004 and 2009 [10]. More than 90% were estimated to be of Swedish origin, and most of the remaining 10% had Finnish background. No information was obtained for controls who were assumed to represent a similar proportion at the time of recruitment 2004–2009. Written informed consent was taken from all the patients. The study was approved by the regional research ethics committees in Stockholm 2002 (Stockholms Regionala Etikprövningsnämnd) and Uppsala (Uppsalas Regionala Etikprövningsnämnd, Dnr: 02-489 and 03-114). Detailed information regarding cancer occurrences in the family comprising of first- and second-degree relatives and cousins was recorded. Based on family history and the pathology and molecular testing for MSI, known cases of FAP and Lynch syndrome were excluded from the study. The selection criteria for the patients to be included in this study was having at least one case of gastric or prostate cancer among close relatives. In total, 685 cases fulfilling these criteria were genotyped and included as cases in this study. Of these 685 CRC cases, 510 were sporadic and 175 were familial. The relatives of these 685 individual CRC cases had various cancers in different locations, with gastric cancer (523) as the most common malignancy, followed by prostate cancer (468) and other various types of cancer, not unique for this sub-cohort of cases. The details regarding the different malignancies among the relatives have been reported previously [12]. Controls were also selected from the Low-risk study consisting of 1642 individuals, of which 536 were spouses of the cases, and 1106 were healthy blood donors from the same geographical region.

### 4.2. Cases for Whole-Genome Sequencing (WGS)

To search for candidate variants in all candidate loci, 89 familial CRC cases with gastric or prostate cancer in their families were used for WGS. The cases were selected based on the maximum number of cases of gastric and/or prostate cancer within the families. The families were described in detail in the previous GWAS with the same cases and different controls [12].

### 4.3. Cases for Whole-Exome Sequencing (WES)

From a previous study on WES in CRC patients, 33 familial CRC cases with gastric and/or prostate cancer in their families were also included to search for candidate variants. Analysis of WES data to search for candidate variants was performed as described [36,37].

### 4.4. Cases and Controls for the Association Studies

For the case-control study of candidate variants, 827 familial CRC cases and 1530 healthy controls were used, as already described. Out of the cases, 691 were from the Low-risk Colorectal cancer study and 136 cases were recruited from the Department of Clinical Genetics at the Karolinska University Hospital, Stockholm [12]. Controls were 990 blood donors from the Stockholm region and 540 healthy spouses from the Low-risk study [14]. In the other case-control study of the same candidate variants, a sub-cohort was selected from the 827 in the first study. In total, 293 cases were selected because they had relatives with gastric and/or prostate cancer. Controls were the same as in the first case-control study.

### 4.5. Genotyping, Quality Control, and Haplotype GWAS

DNA was extracted from blood using the standard protocol in our lab from 2004–2009. The genotyping for both cases and controls was performed at the Center for Inherited Disease Research at Johns Hopkins University, US, using the Illumina Infinium^®^ OncoArray-500K (Illumina, Inc., San Diego, CA, USA) [38]. The first quality control (QC) was done within the CORECT study [13], and a second QC was done at Karolinska Institutet [14]. For the haplotype GWAS analysis, to examine the association between one SNP or haplotype and cancer risk, a logistic regression model was employed. The computational program PLINK V1.07 was used for the analysis and for the calculation of corresponding OR, standard errors, and 95% confidence intervals (CI) [39]. The following parameters were applied while using PLINK v1.07: “hap-logistic” (haplotype logistic regression analysis), “hap-window 1–25” (sliding window sizes 1 to 25), and default settings, which included haplotype phasing with the E-M algorithm, omnibus association test, and minor haplotype frequency of 0.01. Because the number of control persons was smaller than in the previous study, less stringent criteria for *p*-value were used. To find out what *p*-value to use, a hypothetical GWAS using 200 healthy persons was selected from the blood donors and used in a haplotype GWAS where 100 were used as “cases” and 100 as “controls”. No *p*-value was found below 5 × 10^−6^; therefore, a *p*-value criterion of *p* < 5 × 10^−6^ was used for this study. Different sizes of windows were previously tested and windows with more than 25 SNPs rarely gave any results. Thus, a sliding window haplotype analysis of window sizes 1–25 was used. In total, more than eight million sliding windows were tested. In this analysis, each SNP was tested several times, involving 24 SNPs upstream and 24 SNPs downstream. No adjustments were made for gender or age.

### 4.6. Algorithm for Selection of Candidate SNPs Using the Sequencing Data

WGS or WES data from 122 CRC samples was available to search for candidate variants in the candidate risk loci. To be able to search for candidate variants in the sequencing data, GRCh37 base pair positions were converted to GRCh38, as the haplotype data was in genome built 37 and the sequencing data was in genome built 38 (Ensembl genome browser 110). The haplotype region was searched in the Ensembl genome browser to look for genes, non-coding regions, pseudogenes, and regulatory elements. Initially, each chromosome was sorted by base pair positions (smallest to largest). Filters applied were: Synonymous variants, variants without known or rare (<0.005), or common (>0.15) allele frequency were not used. Only single-nucleotide variants (SNV) could be selected for testing with MALDI-TOF (Matrix-Assisted Laser Desorption-Ionization-Time of Flight) in a MassARRAY Platform [40]. The database gnomAD (version v.3.1.2) and the SweGen Variant Frequency database were used for extracting allele frequencies for the variants.

### 4.7. WGS and WES

The Illumina TruSeq PCR-free kit was used for preparing the sequencing libraries, with an average coverage of 30×, according to the manufacturer’s protocol (Illumina, Inc., San Diego, CA, USA). To summarize, Covaris was used for the fragmentation of genomic DNA extracted from 89 peripheral blood samples, and the fragmented DNA was subjected to library preparation involving the end-repair method. This was followed by A-tailing and adaptor ligation. Illumina sequencer NovaSeq6000 was used for the sequencing. Sarek germline pipeline was used to analyze the data [41].

### 4.8. Association Study Using MALDI-TOF

The MALDI-TOF-based MassARRAY platform was used for SNP genotyping analysis. For genotyping, the Core facility at Translational Analysis in Molecular Medicine (TAMM) at the Karolinska University Hospital was utilized [40]. In short, the various steps included primer designing for the various SNPs using a software package from Agena (Agena Bioscience, San Diego, CA, USA). This was followed by PCR amplification of the respective SNP loci, SAP enzyme clean up, extension of the PCR amplicons, and fragment analysis on the Agena MassARRAY analyzer. For automated allele calling, Agena’s SpectroTyper software (https://www.pubcompare.ai/product/uCviCZIBPBHhf-iFVzf3/ (accessed on 14 January 2025)) was used, and validation was done using human DNA from the CEU population genotyped by the HapMap consortium (CEU panel). Positive and negative controls were used in all steps. To ensure the reproducibility of the assay, a few of the samples were repeated. For each individual SNP, association testing was performed separately. Manual calculation was performed for OR using the genotype count in cases and controls; OR > 1 was associated with an increased risk of CRC.

## 5. Conclusions

The findings gave further support to the hypothesis of shared risk between CRC, gastric cancer, and prostate cancer. The ten loci suggested in a previous GWAS were replicated, and 50 new loci, already associated with cancer, were suggested. The results imply potential common genetic pathways involving susceptibility loci across these cancer types. The findings gave further support to the hypothesis of shared risk between CRC, gastric cancer, and prostate cancer. There is still a need for follow-up with new studies to confirm this syndrome and to find other associated tumors. It will be important to find the absolute risks related to these loci before they can be considered for implementation in cancer prevention.

## Figures and Tables

**Table 1 ijms-26-00817-t001:** Comparison of ten loci found in previous GWAS study [12] (first analysis) with current GWAS (second analysis).

Locus	First Analysis BP1–BP2 (GRCh37)	HF	OR	*p*-Value	Genes	Second Analysis BP1–BP2 (GRCh37)	HF	OR	*p*-Value
1q32.2	208968409–209083609	0.03	2.08	3.17 × 10^−8^	No gene	208968409–209083609	0.01	3.35	0.0002
3q29	195750742–195973244	0.02	2.99	2.32 × 10^−8^	*TFRC*, *SLC51A*, *ZDHHC19*, *PCYT1A*	195750742–195973244	0.02	2.14	0.004
4q35.1	185088648–185252818	0.01	2.84	1.30 × 10^−8^	*ENPP6*	185170812–185249147	0.07	1.46	0.001
4q26	119506139–119835148	0.01	3.62	1.59 × 10^−8^	*METTL14*, *SEC24D*, *SYNPO2*	119552849–119809572	0.01	2.28	0.00001
4p15.31	20852244–21112046	0.01	3.25	2.72 × 10^−8^	*KCNIP4*	20864229–21000867	0.01	4.07	4.00 × 10^−4^
8p23.1	11236975–11355821	0.01	3.04	4.47 × 10^−8^	*FAM167A*, *BLK*	11303011–11361552	0.01	2.58	0.0004
13q33.3	109832287–109897922	0.11	1.71	9.20 × 10^−9^	*MYO16*	109796718–109897922	0.03	2.54	21.86 × 10^−6^
13q13.3	37374156–37460648	0.07	1.86	4.15 × 10^−8^	*RFXAP*, *SMAD9*	37366006–37488131	0.05	2.05	8.00 × 10^−6^
16q23.3	82871769–82899877	0.01	3.60	1.38 × 10^−8^	*CDH13*	82866767–82912571	0.02	1.54	0.06
22q11.21	19872009–19930121	0.03	2.30	2.56 × 10^−10^	*TXNRD2*, *COMT*	19889825–19934025	0.09	1.54	9.00 × 10^−5^

HF = Haplotype frequency in samples, OR = Odds ratio. BP1–BP2, first and last position in the haplotype; GRCh37, build 37 of the “human genome”.

**Table 2 ijms-26-00817-t002:** The 55 haplotypes in 50 loci suggested from haplotype GWAS.

Locus	BP1–BP2 (GRCh 37)	HF	OR	*p*-Value	Gene	* PubMed Articles
2p25.2	5593978–5703784	0.02	3.4	3.86 × 10^−6^	no gene	
2p25.1	11503455–11652795	0.07	1.82	3.09 × 10^−6^	*E2F6*, *GREB1*	*E2F6*:153
2q33.1	202688907–202839768	0.02	3.45	3.64 × 10^−7^	*CDK15*	14
2q36.3	226183385–226447735	0.02	3.23	4.32 × 10^−6^	*NYAP2*	1
2q37.3	241894333–241957677	0.04	2.23	2.74 × 10^−6^	*SNED1*	7
4p14	40625135–40677462	0.02	2.4	4.31 × 10^−6^	*RBM47*	46
4q13.1	63992411–64174876	0.03	2.71	3.83 × 10^−6^	no gene	
4q13.3	76049479–76083820	0.02	2.64	5.30 × 10^−7^	no gene	
4q21.22	82401332–82475476	0.01	3.55	2.60 × 10^−6^	*RASGEF1B*	5
4q31.1	141234930–141372158	0.03	2.19	2.81 × 10^−6^	*SCOC, CLGN, MGAT4D*	*SCOC*:16
4q31.3	154605745–154787090	0.02	3.02	3.97 × 10^−6^	*TLR2, RNF175, SFRP2*	*TLR2*:1675; *SFRP2*:575
4q35.1	183942308–184215675	0.01	5.3	3.27 × 10^−6^	*WWC2*	25
5q13.1	67529191–67553636	0.02	3.38	1.05 × 10^−6^	*PIK3R1*	643
6p21.1	45705079–45793972	0.02	3.17	1.09 × 10^−6^	no gene	
6q23.2	133702841–133741558	0.02	3.29	4.54 × 10^−6^	*EYA4*	76
6q25.3	158732602–158973475	0.04	2.03	8.93 × 10^−7^	*TMEM181*, *TULP4*	*TMEM181*:2
6q27	rs2093524-rs10945405	0.01	4.02	2.31 × 10^−6^	*THBS2*	258
7p14.2	36210485–36252293	0.02	4.02	1.62 × 10^−6^	*EEPD1*	19
7p14.1	37298800–37382520	0.01	3.84	1.90 × 10^−6^	*ELMO1*	81
7q22.1	103119863–103130403	0.03	2.72	3.86 ×10^−6^	*RELN*	167
7q35	146559474–146767404	0.01	5.41	3.08 × 10^−7^	*CNTNAP2*	
8p23.2	5430697–5505188	0.02	3.3	1.84 × 10^−6^	no gene	
8q24.22	131997225–132114762	0.02	2.59	4.11 × 10^−6^	no gene	
9p24.3	1124087–1199448	0.05	2.3	2.41 × 10^−8^	no gene	
9p24.3	1876496–1980819	0.02	3.37	7.11 × 10^−7^	no gene	
9q21.31	82433182–82433182	0.61	1.38	2.20 × 10^−6^	no gene	
9q21.31	82445976–82446394	0.54	1.35	3.54 × 10^−6^	no gene	
9q21.33	89168700–89336987	0.01	3.92	9.69 × 10^−7^	no gene	
9q21.33	89481996–89574902	0.05	1.82	4.99 × 10^−6^	*GAS1*	132
9q21.33	89669067–89821219	0.07	1.74	3.83 × 10^−6^	*LINC02872*	1
9q34.2	136822827–136866925	0.11	1.75	1.51 × 10^−6^	*VAV2*	178
10q11.21	45270373–45451178	0.02	3.33	1.33 × 10^−6^	*TMEM72*	
10q26.3	132676113–132742585	0.01	4.77	2.33 × 10^−6^	no gene	
11q14.2	86657520–86836648	0.01	4.34	1.51 × 10^−6^	*FZD4, TMEM135, PRSS23*	*FZD4*:147; *TMEM135*:1
12q24.32	128767084–128916719	0.04	2.46	3.95 × 10^−6^	*TMEM132C*	8
13q12.11	20407151–20686272	0.03	2.52	9.14 × 10^−7^	*ZMYM5, ZMYM2*	*ZMYM5*:3; *ZMYM2*:58
13q12.3	29989466–30075686	0.09	1.77	1.92 × 10^−6^	*MTUS2*	9
13q22.1	74316318–74347673	0.07	1.74	2.29 × 10^−6^	*KLF12*	98
13q33.3	109796718–109897922	0.03	2.54	1.86 × 10^−6^	*MYO16*	12
13q33.3	109886427–109972182	0.04	2.03	4.22 × 10^−6^	no gene	
14q31.3	85180203–85377659	0.02	2.58	2.33 × 10^−6^	no gene	
14q32.11	90422664–90621770	0.04	2.4	3.08 × 10^−6^	*TDP1, KCNK13*	*TDP1*:219
15q13.3	32962642–32976055	0.01	3.52	4.54 × 10^−6^	*SCG5*	65
16p13.3	5420304–5525490	0.01	3.94	3.84 × 10^−6^	*RBFOX1*	
16p13.2	8120348–8277932	0.01	4.3	3.60 × 10^−6^	no gene	
16q23.3	82673410–82691564	0.01	6.35	3.18 × 10^−6^	*CDH13*	308
17p12	14942734–15057691	0.04	2.19	4.31 × 10^−6^	no gene	
17q12	31790250–31888205	0.01	6.52	3.40 × 10^−6^	*ASIC2*	29
17q21.32	46348384–46355550	0.02	2.76	2.46 × 10^−6^	*SKAP1*	40
17q25.3	77925723–77958254	0.01	3.99	4.41 × 10^−6^	*TBC1D16*	11
18q21.2	53179419–53379992	0.03	2.59	2.53 × 10^−7^	*TCF4*	1595
19q12	20411543–20663314	0.02	2.89	4.94 × 10^−6^	no gene	
20q13.33	60966686–60966686	0.71	1.47	2.84 × 10^−7^	*CABLES2*	8
21q21.2	24419862–24555671	0.02	2.86	3.69 × 10^−6^	no gene	
22q12.2	31392606–31438931	0.03	2.41	4.97 × 10^−6^	no gene	

BP1–BP2, first and last position in the haplotype, GRCh37, build 37 of the “human genome”; HF, haplotype frequency in the population; * PubMed articles, the number of publications resulting from a PubMed search using “cancer and gene”.

**Table 3 ijms-26-00817-t003:** Candidate SNPs tested in association analysis of all familial CRC (case-control study) and a sub-cohort of families with colorectal, gastric, and prostate cancer (case-control study).

Locus	SNP	Position (GRCh37)	Gene	Type	Ref Allele	Alt Allele	No of Cases (827)	No Ctrls (1530)	Odds Ratio	*p*-Value	No of Cases (293)	Odds Ratio	*p*-Value
2q33.1	rs34851370	201835676	*CDK15*	missense	C	T	816	1524	1.31	ns	293	1.44	ns
4q31.1	rs358314	140381160	*SCOC*	3’UTR	C	T	730	1341	1.09	ns	263	1.009	ns
	rs358326	140388548	*CLGN*	3’UTR	A	G	817	1525	1.01	ns	292	0.86	ns
	rs72716373	140385611	*SCOC*	3’UTR	T	C	819	1516	0.82	ns	293	0.86	ns
	rs114046582	140398927	*CLGN*	missense	G	A	808	1512	0.87	ns	290	1.08	ns
4q31.3	rs139227237	153703557	*TLR2*	missense	T	C	819	1527	0.95	ns	293	1.20	ns
	rs34803482	153748734	*RNF175*	missense	C	G	805	1434	1.21	ns	289	1.23	ns
	rs115872046	153723390	*RNF175*	missense	G	A	816	1526	1.10	ns	293	1.43	ns
	rs5743708	153705165	*TLR2*	missense	G	A	819	1525	0.97	ns	293	1.16	ns
4q35.1	rs141501417	183260972	*WWC2*	missense	G	A	814	1523	0.77	ns	292	0.53	ns
6q27	rs140852957	169221505	*THBS2*	missense	G	A	818	1522	0.97	ns	293	0.87	ns
10q11.21	rs115943733	44936215	*TMEM72*	3’UTR	G	C	817	1526	1.88	ns	293	3.26	0.009
11q14.2	rs61735304	86954989	*FZD4*	missense	G	A	818	1522	1.50	ns	292	1.19	ns
13q12.11	rs35516773	19993526	*ZMYM2*	missense	G	C	819	1522	1.51	ns	293	0.32	ns
13q12.3	rs1213626528	29505477	*MTUS2*	3’UTR	G	T	819	1523	0	ns	293	0	ns
14q32.11	rs7150480	90043536	*TDP1*	3’UTR	T	C	813	1527	1.23	ns	293	1.16	ns
20q13.33	rs2427307	62391630	*CABLES2*	Intron	G	A	819	1518	0.71	0.0001	293	0.67	0.002

SNP, single nucleotide polymorphism, Position, the exact base pair for the SNP, Positions (GRCh37), build 37 of the “human genome”, Type, type of gene variant, Ref, reference allele, Alt, alternative allele.

## Data Availability

Access to the data is controlled. Variants that fulfilled our selection criteria can be found in the Appendix A. However, Swedish laws and regulations prohibit the release of individual and personally identifying data. Therefore, the whole dataset cannot be made publicly available. The data that support the findings of this study are available from the corresponding authors upon a reasonable request.

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
