# Peer review of "A Haplotype GWAS in Syndromic Familial Colorectal Cancer"

_ijms, 2025, doi:10.3390/ijms26020817_

Round 1

Reviewer 1 Report

Comments and Suggestions for Authors

A Haplotype GWAS in Syndromic Familial Colorectal Cancer, attempts to confirm earlier GWAS findings for CRC associated with familial prostate or gastric cancer.  The study is straight-forward GWAS.  One problem I have is lack of definition of statistical significance levels for each analysis.  In sections 2.1 and 2.2, what are the cut-offs?  I was thinking that 0.001 would be appropriate based on 50 loci.  One feature for the association analysis might be to compare haplotypes between the 293 cases with other cancers in families and the remaining cases with just CRC.  This is not to reveal necessarily significant differences, but to another way to suggest CRC specific vs. syndromic variants in the absence of looking at family members with other cancers.  In connection with this, it is mentioned that haplotypes found were associated with CRC and other cancers, specifically, were any unique? 

Reviewer 2 Report

Comments and Suggestions for Authors

Abstract

The aim should be more clearly identified

Introduction

The format is very strange

Additionally the aim of the study was not clear enough

4. Materials and Methods

4.1. Cases and Controls for GWAS

As the cancer is heredity disease so, the study should take in consideration the nationality of the included patient

Many steps in the method was depend on previously studies so the data is not clear and the author need a new paraphrases to the sentences to be more identified.

4.5. Genotyping, quality control and haplotype GWAS

The extracted and the method of identification is missing

Results Written is good

Discussion seemed as a new result written with no clear discussion

Tables has many abbreviation with no clear ligands

Absence of figures and diagrams explain the study or inclusion and exclusion criteria

References is good

Conclusion is poor with lack correlation with the results and discussion

Round 2

Reviewer 1 Report

Comments and Suggestions for Authors

The authors have addressed my comments satisfactorily. 

Reviewer 2 Report

Comments and Suggestions for Authors

The authors succeeded in answering all the reviewer commends